# Importance of the COVID-19 Vaccine Booster Dose in Protection and Immunity

**DOI:** 10.3390/vaccines10101708

**Published:** 2022-10-13

**Authors:** Alireza Abdollahi, Yeganeh Afsharyzad, Atefeh Vaezi, Alipasha Meysamie

**Affiliations:** 1Department of Pathology, School of Medicine, Imam Hospital Complex, Tehran University of Medical Sciences, Tehran 1416634793, Iran; 2Department of Microbiology, Faculty of Modern Sciences, The Islamic Azad University of Tehran Medical Sciences, Tehran 1416634793, Iran; 3Cancer Prevention Research Center, Isfahan University of Medical Sciences, Isfahan 8174673461, Iran; 4Department of Community Medicine, School of Medicine, Tehran University of Medical Sciences, Tehran 1416634793, Iran; 5Community-Based Participatory Research Center, Iranian Institute for Reduction of High Risk Behaviors, Tehran University of Medical Sciences, Tehran 1416634793, Iran

**Keywords:** SARS-CoV-2, vaccination, booster dose, RBD antibody, neutralizing antibody, spike antibody

## Abstract

Background: There is debate on the necessity of booster doses of COVID-19 vaccination, especially in countries with limited resources. Methods: This cross-sectional study was conducted in a referral laboratory in Tehran, Iran. The level of COVID-19 antibodies was measured and compared between individuals regarding the number of COVID-19 vaccine shots. Results: In this study, 176 individuals with a mean age of 36.3 (±11.7) years participated. A total of 112 individuals received two doses of the COVID-19 vaccine, and 64 individuals received three doses. Level of all antibodies was higher in those who received three doses than in those who received two doses of the COVID-19 vaccine. Considering the SARS-CoV-2 Spike IgG, the difference was not statistically significant but for the SARS-CoV-2 RBD IgG and SARS-CoV-2 NAB the difference was statistically significant. Regarding to the background variables, receiving influenza vaccine in the past year, history of autoimmune diseases and past medical history of chicken pox showed a significant association with the number of vaccine doses received. Their effects on the outcome variables assessed with multivariate logistic regression analysis. Conclusion: The results of our study show that a booster dose of the COVID-19 vaccine enhances the antibody response.

## 1. Introduction

The emergence of SARS-COV-2 in 2019 resulted in the COVID-19 pandemic and still involves countries [1]. About 30 months after the first case of COVID-19, about 551 million cases and 6.3 million deaths were reported globally [2]. Vaccination is the main promising strategy to return to everyday life in the fight against COVID-19. The level of vaccination varies in different countries. Some countries have begun to implement third and fourth doses, while others are still lagging behind. In Iran, 150 million doses of COVID-19 vaccines were administered by July 2022; about 77% of the population had at least one shot, 68.8% of the population was fully vaccinated, and only 32.7% received their booster dose [3].

COVID-19 vaccines have different platforms with different mechanisms of action, including nucleic acid, protein-based, viral-vector-based, inactivated virus, and live-attenuated virus platforms [4]. Vaccine-induced immunity against viral infections mainly relies upon neutralizing antibodies (NABs) along with cellular immunity induced by T-lymphocytes [5]. The fusion of the virus to the host cell is caused by the receptor binding protein (RBD), which is a part of the structure of the virus spike glycoprotein and is a good target for improving NABs [6].

All the approved COVID-19 vaccines reported to have an acceptable increase in the level of RBD-binding IgG and neutralizing antibodies in their phase I/II trials. In the trial of BNT162b1, geometric mean of NAB titer increased by 1.9–4.6-fold compared to COVID-19-infected patients; for RBD-binding IgG, the same increase was also documented [7]. ChAdOx1-S enhances the spike-specific T cell response and this enhancement is boosted after the second dose [8]. Anti-spike antibody increased after the first dose of mRNA-1273 and improved after the second dose; two months after the second dose, NAB was detected in all participants [9]. Seroconversion and elevated antibody levels were reported in more than 95% of participants in phases I and II of the BBIBP-CorV vaccine [10]. Regarding the rAd5-S/rAd26-S vaccine, seroconversion and cell-mediated response were detected in 100% of the participants [11].

From the public health perspective, herd immunity is the ultimate goal [12]. Still, there are some challenges, including the antigenic evolution of SARS-CoV-2 to escape immunity [13], the decrease in the level of infection/vaccine-induced protective antibodies over time [14,15,16], and also the necessity of booster doses regarding populations with limited access even to the first dose [17,18]. Hence, governments need to decide whether to mobilize the population to get booster shots or not. In this study, we tried to investigate the humoral response induced by the COVID-19 booster dose (regardless of the vaccine type) to help policymakers make evidence-informed decisions.

## 2. Materials and Methods

### 2.1. Study Design and Participants

This cross-sectional study was conducted in January 2022 in the laboratory of Imam Khomeini hospital, which is a referral hospital affiliated with the Tehran University of Medical Sciences, Tehran, Iran. Among those who were referred to the laboratory for any reason, COVID-19-vaccinated adults were enrolled using a convenience sampling method. This study was approved by the Ethics Committee of the Tehran University of Medical Sciences (IR.TUMS.Medicine.REC.1400.1297). Participants were informed about the aims of the study, and all voluntarily accepted the participation. All data were managed, analyzed, and reported anonymously, and all COVID-19 Ab tests were performed free of charge for the participants. The Strengthening the Reporting of Observational Studies in Epidemiology (STROBE) guideline was used to report the study results [19].

### 2.2. Demographics

Data on demographic variables such as age, gender (female, male), height, weight, body mass index (BMI), type of received vaccine for the first, second, and booster dose (BBIBP-CorV, rAd26-S/rAd5-S, Bharat Covaxin, ChAdOx1-S, BNT162b2, and CoVIran Barekat), comorbidities (mental health problems, cardiovascular diseases, respiratory disorders, autoimmune diseases, anemia, and diabetes), medications, history of previous infections (measles, influenza, and chicken pox), history of COVID-19 infection, and administration of influenza vaccine during the last year were gathered. Two groups were formed based on the number of COVID-19 vaccines doses received: the first group included individuals who had received two doses, and the second group who had received three doses.

### 2.3. Measurement of Antibodies

A venous blood sample of 5 mL was collected into an ethylenediaminetetraacetic acid-coated microtainer from all participants. Samples were centrifuged, and the serum was separated. The serum levels of SARS-CoV-2 spike IgG, SARS-CoV-2 RBD IgG, and SARS-CoV-2 NAB were evaluated using Pishtaz Teb-specific ELISA kits. The results were interpreted according to the references provided by the Pishtaz Teb company.

Pishtaz Teb ELISA kit (product code: PT-SARS-CoV-2 Spike IgG-96) was used to measure the level of SARS-CoV-2 spike IgG in the serum. The specificity and sensitivity of the kit are 99.01% and 98.16%, respectively. A concentration of ≥8 relative units per milliliter (RU/mL) is assumed as positive.

To measure the concentration of SARS-CoV-2 RBD IgG in blood samples, we used the Pishtaz Teb ELISA kit (product code: PT-SARS-CoV-2-RBD-IgG-96) with a specificity of 100% and a sensitivity of 97.1%. The concentration of ≥5 RU/mL is assumed to be positive.

For SARS-CoV-2 NAB, the level of NAB assessed using the Pishtaz Teb ELISA kit (product code: PT-SARS-CoV-2-Neutralizing-Ab-96). Based on the product brochure, antibody levels ≥ 2.5µg/mL are supposed to be a positive result.

### 2.4. Data Analysis

Data were analyzed using SPSS version 25 (IBM Corp., Armonk, NY, USA). A mean and standard deviation were used to describe continuous variables. A frequency table was used to present categorical variables. To compare continuous and categorical variables between groups, independent sample *t*-tests and chi-squared tests were used, respectively. Variables with statistically significant associations in univariate analysis were included in logistic regression analysis to control the confounding effects of background variables. The level of significance was set below 0.05.

## 3. Results

In this cross-sectional study, 176 individuals (119(67.6%) female, 57(32.4%) male) who were referred to the laboratory of Imam Khomeini hospital were assessed. The mean (±SD) age of the study population was 36.3 (±11.7) years. The frequency of comorbidities was almost low in the study sample. Only four (2.3%) and two (1.1%) individuals reported cardiovascular and respiratory diseases, respectively and no one had a history of diabetes mellitus. Any history of an autoimmune disease was documented in three (1.7%) individuals. A history of COVID-19 infection was documented in 57.2% of the study population.

Considering COVID-19 vaccination, 112 individuals (63.6%) received two doses of the COVID-19 vaccine of any kind, and 64 individuals (36.4%) received three doses. Most of the study population (107(60.8%)) received BBIBP-CorV for the first dose. Other vaccines include BNT162b2 (1(0.6%)), ChAdOx1-S (12(6.8%)), rAd26-S/rAd5-S (25(14.2%)), CoVIran Barekat (10(5.7%)), and Bharat Covaxin (21(11.9%)). For the second dose, 108(62.4%) received BBIBP-CorV, 1(0.6%) received BNT162b2, 11(6.4%) received ChAdOx1-S, 24(13.9%) received rAd26-S/rAd5-S, 10(5.8%) received CoVIran Barekat, and 19(11.0%) received Bharat Covaxin. For the third dose, ten individuals (16.4%) received BBIBP-CorV, 51(83.6%) received ChAdOx1-S. Details are presented in Appendix A.

Based on the number of the COVID-19 vaccine doses received, the sample population was categorized into two groups of two or three doses. The mean±SD of age was not statistically different between the two groups (36.8 ± 12.1 vs. 35.6 ± 11.0, *p*-value, 0.504). The frequency of the female gender was 64.3% and 73.4% in the groups that received two and three doses, respectively (*p*-value, 0.212). There was no statistically significant difference in the distribution of comorbidities between the two groups except for autoimmune diseases. All three individuals with an autoimmune disease received three doses of the COVID-19 vaccine. Past medical history of COVID-19 infection was reported in 59 (52.6%) individuals with two doses and 40 (62.5%) individuals who received three doses (*p*-value, 0.207). Past medical history of chicken pox and history of influenza vaccination in the past year differed between the two groups (*p*-value, 0.004 and <0.001, respectively). Other variables are presented in Table 1.

The serum levels of SARS-CoV-2 spike IgG, SARS-CoV-2 RBD IgG, and SARS-CoV-2 NAB were evaluated (Table 2 and Table 3). More than 90% of the study population had positive levels of SARS-CoV-2 spike IgG (mean± SD, 68.5 ± 32.5). The frequency of positive spike IgG was higher in individuals who received three doses than in individuals with two doses, but it was not statistically significant (96.9% vs. 90.2%, respectively; OR, 3.4, 95% CI, 0.7–15.6; *p*-value, 0.137). Meanwhile, the level of spike IgG was higher in those who received three doses than in those who received two doses (82.8 ± 24.2 vs. 60.4 ± 33.8, respectively; *p*-value < 0.001).

The frequency of positive levels of RBD IgG and NAB was 81.3% and 75.0% in the whole sample, respectively. Administration of the third dose increased the frequency of the protective rate considering RBD IgG (OR, 7.5; 95% CI, 2.1–25.6; *p*-value < 0.001) and NAB (OR, 8.3; 95% CI, 2.8–24.3; *p*-value < 0.001). RBD AB increased significantly with the third dose of the COVID-19 vaccine (from 28.2 ± 25.6 to 51.0 ± 29.7, *p*-value < 0.001). The increase in NAB was also notable, from 23.9 ± 27.5 to 51.3 ± 24.1 (*p*-value < 0.001). The serum level of SARS-CoV-2 Spike IgG, SARS-CoV-2 RBD IgG, and SARS-CoV-2 NAB are presented in Figure 1.

To control the confounding variables in the prediction of positive levels of antibodies, age, gender, BMI, past medical history of autoimmune disease, past medical history of chicken pox, and past medical history of influenza vaccination, along with the number of COVID-19 vaccine dosages, have been entered into three separate logistic regression models. The result of the logistic regression for each antibody is presented in Table 4.

In the logistic regression for the prediction of positive anti-spike antibody (Cox and Snell R squared, 0.1), as in the univariate analysis, the number of vaccine dosages was not associated with positive anti-spike antibody levels in the study population (*p*-value, 0.066). Moreover, the association between age (OR, 1.08; 95% CI, 1.02–1.14; *p*-value, 0.005) and BMI (OR, 0.15; 95% CI, 0.02–0.84; *p*-value, 0.031) with the positive level of SARS-CoV-2 Spike IgG was statistically significant.

The positive level of SARS-CoV-2 RBD IgG showed a positive association with the number of COVID-19 vaccine dosages (OR, 2.27; 95%CI, 2.95–166.66; *p*-value, 0.003) and a negative association with a medical history of autoimmune disease (OR, 0.03; 95% CI, 0.00–0.77; *p*-value, 0.034) (Cox and Snell R squared, 0.134).

The association between the number of dosages with the frequency of protective SARS-CoV-2 NAB levels remained statistically significant (OR, 16.66; 95% CI, 3.84–100.00; *p*-value < 0.001) in multiple logistic regression analysis. Other variables showed no significant association (Cox and Snell R squared, 0.16).

## 4. Discussion

Although we are in the third year of the COVID-19 pandemic, there are still unanswered questions, especially regarding the COVID-19 vaccines. The level of protective antibodies, the role of heterologous vaccine regimens, and optimum dose intervals are still under question. A higher titer of antibodies is supposed to be associated with more extended and higher protection, especially against emerging COVID-19 variants [20,21]. No matter the type of COVID-19 vaccine administered, the waning of humoral responses is observed, especially in older adults, immunosuppressed individuals, and males [22,23]. Meanwhile, the willingness to receive a booster dose has decreased in communities, and “doubt on the necessity of further vaccination” is mentioned as the main reason [24,25,26]. 

The vaccine-induced immune response is strongly affected by host factors (age, gender, genetics, history of COVID-19 infection, and comorbidities) and vaccine factors (vaccine type, adjuvants, number of doses, and vaccination schedule) [27,28,29]. We examined the relationships of different factors as determinants of vaccine response, including age, gender, BMI, comorbidities, medications, history of measles, influenza, and chicken pox, history of COVID-19, administration of influenza vaccine in the past year, and the number of vaccine doses.

The results of our study, in line with other studies, showed that the number of doses is a significant determinant of antibody concentration [30,31,32]. Our study indicated that the level of SARS-CoV-2 spike IgG, SARS-CoV-2 RBD IgG, and SARS-CoV-2 NAB were significantly higher after the booster dose with about a 1.5–2-fold increase in their titer. 

Age is a determinant of immunity response, as the production of antibodies decreases with age due to impairment in T-cells and maturation of B-cells [33]. However, the only antibody associated with age in our study was the SARS-CoV-2 Spike IgG. In contrast to our study, in a survey by Uysal et al., age had no statistically significant relationship with the titer of RBD antibody [34]. This contradiction could be explained by the time of antibody measurement, as the antibody level decreases over time, which could be apart from the effect of age on the antibody level. The results of a study by Levin et al. show that adults over 65 years old have lower levels of antibodies compared to younger adults [22]. In this study, the mean age of the participants was around 36 years old, and older adults did not participate, which could explain the difference in the association of age with the frequency of protective antibodies in our study with previous studies.

Immunosuppression is a determinant of antibody concentration after vaccination. The results of two studies by Boyarski et al. show an increase in immunity after the second dose in organ transplant receivers. Immunity is detected in 15% and 54% after the first and second doses, respectively [35,36]. The result of our study was in line with previous studies. In our study, having an autoimmune disease is negatively associated with positive levels of RBD IgG. In contrast, administering the third dose was positively associated with positive levels of RBD IgG. In organ transplant recipients and cancer patients, the levels of antibodies were boosted after the third dose [37,38]. In a study on a group of solid organ transplant recipients, about half of those who were seronegative after the second dose became seropositive after the third dose [39]. It has also been indicated that the odds of having a positive test and hospitalization decrease after three doses of the BNT162B2 vaccine compared to two doses [40].

Another determinant of vaccine-induced immunity is obesity, which is negatively associated with antibody concentration, and obese individuals are more at risk of breakthrough COVID-19 infection [41]. Individuals with a BMI of 25 or higher had a lower likelihood of having positive SARS-CoV-2 spike Ab.

There are some limitations to acknowledge in our study. First, we did not measure the period between the vaccine shot and antibody evaluation, so we cannot make a definite conclusion regarding the titer of antibodies. Second, due to our limited sample size, this study has no claim on the effectiveness of different vaccines. Third, we aim to investigate the level of humoral response at a particular point in time. Still, the lack of follow-up data, especially on clinical outcomes, could be mentioned as another limitation, and longitudinal studies are needed. Finally, the power of our models is not high enough. Therefore, there are other factors influencing the level of antibodies that are not included in our study.

## 5. Conclusions

Different variables are associated with the titer of protective antibodies induced by COVID-19 vaccines. In conclusion, the results of our study show that the booster dose of the COVID-19 vaccine is a strong determinant of positive SARS-CoV-2 RBD IgG and SARS-CoV-2 NAB, which best correlates with immunity.

## Figures and Tables

**Figure 1 vaccines-10-01708-f001:**
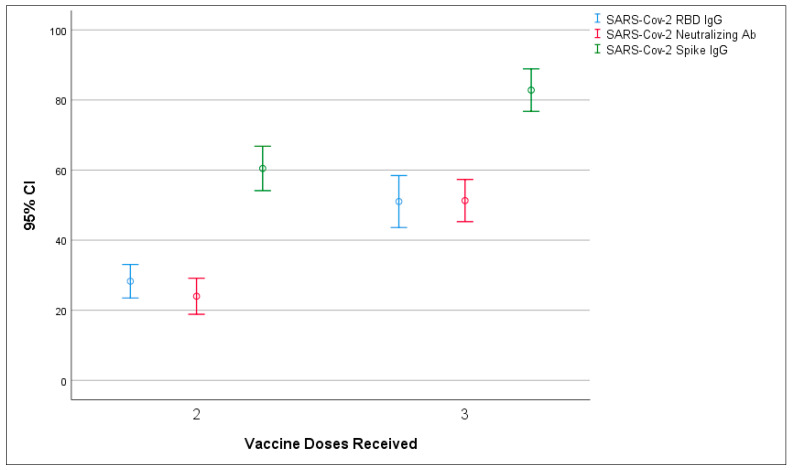
Comparison of the serum level of antibodies in twice- and triple-vaccinated individuals.

**Table 1 vaccines-10-01708-t001:** Baseline variables of the study population regarding the number of vaccine doses received.

	Whole Sample N = 176	Receiving Two Doses N = 112	Receiving Three Doses N = 64	*p*-Value
Age (mean ± SD)	36.3 ± 11.7	36.8 ± 12.1	35.6 ± 11.0	0.504
Gender				0.212
Female	119 (67.6)	72 (64.3)	47 (73.4)	
Male	57 (32.4)	40 (35.7)	17 (26.6)	
BMI (mean ± SD)	25.0 ± 7.7	24.6 ± 2.8	25.6 ± 11.9	0.428
<25	91 (56.5)	54 (55.1)	37 (58.7)	0.650
≥25	70 (43.5)	44 (44.9)	26 (41.3)	
Comorbidities				
Mental health problems	2 (1.1)	0	2 (3.1)	0.131
Cardiovascular diseases	4 (2.3)	3 (2.6)	1 (1.5)	1.000
Respiratory disorders	2 (1.1)	1 (0.8)	1 (1.5)	1.000
Autoimmune diseases	3 (1.7)	0	3 (4.6)	0.047
Anemia	29 (16.5)	19 (16.9)	10 (15.6)	0.798
Medications				
Iron	29 (16.5)	18 (16.0)	11 (17.1)	0.848
Folic acid	25 (14.2)	16 (14.2)	9 (14.0)	0.967
Supplements	28 (16.1)	16 (14.2)	12 (18.7)	0.467
Anti-hypertensive	12 (6.9)	11 (9.8)	1 (1.5)	0.058
Past medical history of measles	13 (7.6)	8 (7.1)	5	0.850
Past medical history of influenza	29 (16.8)	16 (14.2)	13 (20.3)	0.302
Past medical history of chicken pox	54 (31.2)	26 (23.2)	28 (43.7)	0.004
History of COVID-19 infection	99 (56.3)	59 (52.7)	40 (62.5)	0.207
COVID-19 infection frequency				0.368
Once	75 (76.5)	47 (81.0)	28 (70.0)	
Twice	18 (18.4)	8 (13.8)	10 (25.0)	
Three times	5 (5.1)	3 (5.2)	2 (5.0)	
Receiving influenza vaccine in the past year	45 (25.9)	18 (16.0)	27 (42.1)	<0.001

Data is presented in number (percent) unless otherwise stated. BMI: body mass index; SD: standard deviation.

**Table 2 vaccines-10-01708-t002:** Frequency of immune level of antibodies in groups of 2 and 3 doses of COVID-19 vaccine.

	Whole Sample N = 176	Receiving Two Doses N = 112	Receiving Three Doses N = 64	OR (95% CI)	*p*-Value
SARS-CoV-2 spike IgG				3.37 (0.72–15.62)	0.137
Positive	163 (92.6)	101 (90.2)	62 (96.9)		
Negative	13 (7.4)	11 (9.8)	2 (3.1)		
SARS-CoV-2 RBD IgG				7.46 (2.16–25.64)	<0.001
Positive	143 (81.3)	82 (73.2)	61 (95.3)		
Negative	33 (18.8)	30 (26.8)	3 (4.7)		
SARS-CoV-2 NAB				8.33 (2.81–24.39)	<0.001
Positive	132 (75.0)	72 (64.3)	60 (93.8)		
Negative	44 (25.0)	40 (35.7)	4 (6.3)		

Data are presented in numbers (percentage). IgG: immunoglobulin G; NAB: neutralizing antibody; RBD: receptor binding domain.

**Table 3 vaccines-10-01708-t003:** The level of anti-SARS-CoV-2 antibodies regarding the number of received vaccine doses.

	Whole Sample N = 176	Receiving Two Doses N = 112	Receiving Three Doses N = 64	*p*-Value
SARS-CoV-2 spike IgG (RU/mL)	68.5 ± 32.5	60.4 ± 33.8	82.8 ± 24.2	<0.001
SARS-CoV-2 RBD IgG (RU/mL)	36.5 ± 29.2	28.2 ± 25.6	51.0 ± 29.7	<0.001
SARS-CoV-2 NAB (µg/mL)	33.9 ± 29.4	23.9 ± 27.5	51.3 ± 24.1	<0.001

Data are presented as mean ± SD. IgG: immunoglobulin G; NAB: neutralizing antibody; RBD: receptor binding domain; RU/mL: relative units per milliliter; µg/mL: micrograms per milliliter.

**Table 4 vaccines-10-01708-t004:** Logistic regression analysis predicting the positivity of antibodies.

Type of Antibody	Variables	B	S.E.	OR	*p*-Value	95% CI for OR	
						Lower	Upper
SARS-CoV-2 spike IgG	Number of dosages *	1.98	1.08	7.69	0.066	0.87	100.00
	Age	0.08	0.02	1.08	0.005	1.02	1.14
	BMI ***	−1.84	0.85	0.15	0.031	0.02	0.84
SARS-CoV-2 RBD IgG	Number of dosages *	3.11	1.03	2.27	0.003	2.95	166.66
	Autoimmune disease **	−3.36	1.58	0.03	0.034	0.00	0.77
SARS-CoV-2 NAB	Number of dosages *	2.80	0.75	16.66	0.000	3.84	100.00

* The reference is receiving two doses of COVID-19 vaccine. ** The reference is a negative history. *** The reference is BMI < 25. IgG: immunoglobulin G; RBD: receptor binding domain; NAB: neutralizing antibody; BMI: body mass index.

## Data Availability

All relevant data are available in the article.

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
