# Peer review of "Importance of the COVID-19 Vaccine Booster Dose in Protection and Immunity"

_vaccines, 2022, doi:10.3390/vaccines10101708_

Round 1

Reviewer 1 Report

Authors of the study describe impact of receiving a booster dose on anti-SARS-CoV-2 antibody levels in 176 individuals. Overall the study is well deigned and benefits from  a well described patient characteristics.

Major comment:

Could the authors elaborate on why didn’t they perform the analysis of impact of the type of vaccine received? For the BNT162b2 there is just one participant so no conclusions can be drawn, but for the others, a small comparison could be made. This seems like a lost opportunity.

The article would benefit for a graph comparing measured antibody levels in twice and three times vaccinated individuals.

Major comment.

Can the authors explain the P values of 0.000 in Table 3?

Minor comment:

In table 3 the unit in which antibody levels was measured should be indicated.

Some remarks regarding wording:

On several occasions the authors refer to SARS-CoV-2 specific antibodies as “protective”, while this term should be used with caution. Not every antibody induced by infection or vaccination is protective. I suggest reserving the term to NAB ELISA results.

Line 44

viral-based vector platforms

Should be viral-vector based platforms instead

124-125

The frequency of comorbidities was almost  as low in the study sample.

The sentence needs clarification. What do the authors mean by “as low”.

Author Response

Importance of the Covid-19 vaccine booster dose in protection and immunity

Reviewer’s comments

Reply

1

Could the authors elaborate on why didn’t they perform the analysis of impact of the type of vaccine received? For the BNT162b2 there is just one participant so no conclusions can be drawn, but for the others, a small comparison could be made. This seems like a lost opportunity.

Thanks for your comment; in our population, the usage of Covid-19 vaccines in the first, second, and third doses was very heterogeneous, and we had different compositions with a small number of samples in each group. Hence we decided to provide an overall view in this article in order to have a general report with a public health perspective.

2

The article would benefit for a graph comparing measured antibody levels in twice and three times vaccinated individuals.

Thanks for your comment, we add a graph

3

Can the authors explain the P values of 0.000 in Table 3?

We mean <0.001

Edited

4

In table 3 the unit in which antibody levels was measured should be indicated.

Edited

The unit is added

5

On several occasions the authors refer to SARS-CoV-2 specific antibodies as “protective”, while this term should be used with caution. Not every antibody induced by infection or vaccination is protective. I suggest reserving the term to NAB ELISA results.

We edited different sentences.

6

Line 44

viral-based vector platforms

Should be viral-vector based platforms instead

Edited

7

124-125

The frequency of comorbidities was almost  as low in the study sample.

The sentence needs clarification. What do the authors mean by “as low”.

Edited

The frequency of comorbidities was almost low in the study sample.

Reviewer 2 Report

The manuscript " Importance of the Covid-19 vaccine booster dose in protection 2 and immunity", that aimed to study different variables associated with the titer of protective antibodies induced by different COVID-19 vaccines, and the results of this study showed that the booster dose of 259 of the Covid-19 vaccine is a strong determinant of protective SARS-Cov-2 RBD IgG and 260 SARS-CoV-2 NAB, which best correlates with immunity. The methodology, result ,and discussion sections were written obviously.

Author Response

dear reviewer

Thank you for your comments; we edited and improved the methods and results sections